# AQP4 Attenuated TRAF6/NFκB Activation in Acrylamide-Induced Neurotoxicity

**DOI:** 10.3390/molecules27031066

**Published:** 2022-02-04

**Authors:** Chia-Yu Hung, Chih-Han Chang, Tzu-Jung Lin, Hsin-Hui Yi, Nian-Zhen Tsai, Yu-Ru Chen, Yng-Tay Chen

**Affiliations:** 1Graduate Institute of Food Safety, College of Agriculture and Natural Resources, National Chung Hsing University, Taichung 402, Taiwan; g109050022@mail.nchu.edu.tw (C.-Y.H.); g109050018@mail.nchu.edu.tw (T.-J.L.); g109050006@mail.nchu.edu.tw (H.-H.Y.); g110050010@mail.nchu.edu.tw (N.-Z.T.); g110050016@dragon.nchu.edu.tw (Y.-R.C.); 2Bachelor Program of Biotechnology, College of Agriculture and Natural Resources, National Chung Hsing University, Taichung 402, Taiwan; s107030313@mail.nchu.edu.tw

**Keywords:** acrylamide, neurotoxicity, depression, MMP-9, AQP4, TRAF6

## Abstract

Acrylamide (ACR) is present in high-temperature-processed high-carbohydrate foods, cigarette smoke, and industrial pollution. Chronic exposure to ACR may induce neurotoxicity from reactive oxygen species (ROS); however, the mechanisms underlying ACR-induced neurotoxicity remain unclear. We studied 28-day subacute ACR toxicity by repeatedly feeding ACR (0, 15, or 30 mg/kg) to rats. We conducted RNA sequencing and Western blot analyses to identify differences in mRNA expression in the blood and in protein expression in the brain tissues, respectively, of the rats. AQP4 transient transfection was performed to identify potential associations with protein regulation. The rats treated with 30 mg/kg ACR exhibited hind-limb muscle weakness. Matrix metalloproteinase (MMP9) expression was higher in the ACR-treated group than in the control group. ACR induced MMP-9 and AQP4 protein expression in the brain tissues of the rats, which subsequently presented with neurotoxicity. In the in vitro study, Neuro-2a cells were transiently transfected with AQP4, which inhibited MMP-9 and TNF receptor-associated factor 6 (TRAF6) expression, and inhibited ACR induced expression of TRAF6, IκBα, and nuclear factor κB (NFκB). Using a combination of in vivo and in vitro experiments, this study revealed that depressive symptoms associated with ACR-induced neurotoxicity are associated with downregulation of AQP4 and induction of the TRAF6 pathway.

## 1. Introduction

Acrylamide (ACR) is a water-soluble unsaturated amide mainly used to produce polyacrylamide for industrial purposes [1]. ACR is also present in cigarette smoke and carbohydrate-rich food ingredients processed by high-temperature frying or baking, such as fries, potato chips, coffee, and bread, as a product of Maillard reactions [2]. Due to its neurotoxicity [3], reproductive toxicity [4], genotoxicity, and carcinogenicity to animals [5], ACR has been classified as “reasonably anticipated to be a human carcinogen” by the National Toxicology Program and as a group 2A carcinogen by the International Agency for Research on Cancer [6]. Among the types of ACR-induced toxicity, neurotoxicity is the earliest to present and has been studied the most thoroughly. Several studies have reported that ACR-induced oxidative stress may contribute to neurotoxicity; common symptoms exhibited by rats exposed to ACR have included hind-limb foot splay, ataxia, and skeletal muscle weakness [7]. The early symptoms exhibited by workers exposed to ACR are skin peeling on the hands, impairment of vibration sensation in the toes, weakness in the leg muscles, and numbness in the hands and legs; workers with long-term ACR exposure experience limb weakness, anorexia, truncal ataxia, and horizontal nystagmus [8].

Depression, also known as major depressive disorder (MDD) or clinical depression, is a mood disorder that causes persistent feelings of sadness and loss of interest. MDD affects how an individual feel, thinks, and behaves and can lead to various emotional and physical problems. Individuals with depression may have trouble engaging in normal day-to-day activities and sometimes feel as though life is not worth living. A study reported that ACR–hemoglobin adduct levels are associated with depressive symptoms in US adults [9]. MDD is a chronic mental disorder associated with symptoms including depressed moods, loss of interest in activities, loss of energy and thinking ability, feelings of fatigue and worthlessness, and suicidal ideations [10,11]. Although MDD is common, the detailed mechanisms of its pathogenesis are not fully understood. Studies have revealed that immune dysregulation and long-term neuroinflammation, both of which might be triggered by stress responses, are positively correlated with MDD [12]. Excess proinflammatory cytokines have been reported to induce numerous neurotoxic effects in the brain, such as reducing neurogenesis in the hippocampus and neuronal cell proliferation and releasing glutamate to induce cell excitement [13]. Therefore, inflammation may be a key factor of neurotoxic effects that may result in MDD. Matrix metalloproteinase (MMP-9) expression is positively correlated with the severity of depression in patients with MDD; elevated MMP-9 activity has been observed in the hippocampi of in animal models of depression [14,15], suggesting that MMP-9 is a potential biomarker in the pathology of MDD aquaporin 4 (AQP4), a water-selective channel protein widely expressed in the central nervous system (CNS), also contributes to the pathology of MDD. In a postmortem study, patients with depression exhibited lower AQP4 mRNA and protein expression in the brain than healthy control [16,17]. AQP4 knockout also revealed negative effects on neuroprotective markers and behavioral outcomes and was associated with increases in inflammatory markers in animal models of depression [18], indicating that AQP4 exhibits neuroprotective functions in the CNS and against MDD. MMP-9, a proteolytic enzyme related to synaptic plasticity in the brain activates proinflammatory cytokines in neuroinflammation and is released by T cells during immunoreactions [19]. The abundant expression of Toll-like receptor (TLR) 4 induces microglial activation, triggering the tumor necrosis factor receptor-associated factor 6 (TRAF6) [20]. Thus, inhibiting microglial activation is crucial in treating neuroinflammation-associated diseases, including depression.

As the ACR-regulated genetic pathways and molecular mechanisms of depressive symptoms remain poorly understood, we investigated the roles of neurotoxicity-related genes—namely, *AQP4*, *MMP-9*, and *TRAF6*, in an in vivo rat model and in vitro cell models of ACR-induced neurotoxicity.

## 2. Results

### 2.1. Acrylamide-Induced Neurotoxicity Symptoms in Rats

We studied 28-day subacute ACR toxicity by repeatedly feeding ACR (0, 15, or 30 mg/kg) to rats. Rats treated with 15 mg/kg ACR showed hindlimb and some difficulty in walking at day 28. Additionally, rats treated with 30 mg/kg ACR exhibited hindlimb muscle weakness, difficulty in walking, limb numbness, decreased tactile sensation, and disappearance of tendon reflexes, indicating that ACR induced neurotoxicity in the rats. The pathological exam results revealed the normal morphologies of neurons in the molecular, Purkinje cell, and granular layers in the control group (Figure 1). The Purkinje cells of the rats in the 15 mg/kg and 30 mg/kg ACR groups exhibited abnormal morphologies; the Purkinje cells in the cerebellar cortices of the rats had irregular shapes and elevated basophilic chromatin, which manifested as deep-stained nuclei with no visible nucleoli. The surrounding tissue gap became wider. These results indicate ACR caused neurotoxicity in rats.

### 2.2. ACR Alters Genes Expression in Rats

We sought to evaluate the gene regulation that is associated with ACR-induced neurotoxicity. The rats were administered oral 0 mg/kg (control), 15 mg/kg, or 30 mg/kg ACR for 28 days, and their blood samples were tested for differences in gene expression. Compared with the control group, the expression of 120 and 570 genes were inhibited and induced, respectively, in the 15 mg/kg ACR group (Figure 2A), and the expression of 440 and 520 genes were inhibited and induced, respectively, in the 30 mg/kg ACR group (Figure 2B). Compared with the control group, a total of 189 and 919 unique genes were inhibited and induced in both ACR dosing groups (Figure 2C). According to the Kyoto Encyclopedia of Genes and Genomes (KEGG) public pathway database, there were six pathways difference between control and ACR-treated groups—namely, organismal systems, metabolism, human disease, genetic information processing, environmental information processing, and cellular processes (Figure 3). The numbers 27, 14, 7, and 6 of the identified genes are related to genetic information processing: ubiquitin-mediated proteolysis, Fanconi anemia pathway, and homologous recombination, respectively (Figure 3). As ACR-induced neurotoxicity manifests as depressive symptoms, we investigated the expression of MMP-9, AQP4, and TRAF6 mRNA. The logFC values of MMP-9, TRAF6, NFκB, IκBα, and heme oxygenase-1 (HO-1) expression were 1.65, 1.23, 0.14, 0.32, and 0.71, respectively (Table 1, Figure 4), but the expression of AQP4 was not detected. These phenomena indicate ACR may induce depressive symptoms.

### 2.3. ACR Induces MMP-9 and AQP4 Expression in the Brain Tissue

According to the blood gene results, we detected the expression of MMP-9 and AQP4 in the brain tissues of the rats (Figure 5). The 28-day ACR treatment-induced MMP-9 overexpression in the 30 mg/kg group and dose-dependent increased in TRAF6, IκBα, NFκB, nNOS, and nAChR expression (Figure 5). However, ACR inhibited AQP4 expression in the 30 mg/kg group. Results showed ACR-induced depressive symptoms are associated with protein MMP-9 activation and AQP4 inhibition.

### 2.4. AQP4 Regulates ACR-Induced MMP-9 Expression

To investigate the role of AQP4 in ACR-induced depressive symptoms, we constructed a recombinant vector containing rat AQP4 cDNA (pcDNA3.0-AQP4), to conduct in vitro transient transfection of Neuro-2a cells. The results indicated that the overexpression of AQP4 inhibited MMP-9, TRAF6, and NFκB expression (Figure 6) but did not affect nNOS and nAChR expression. ACR treated with AQP4 overexpression cells showed inhibition of TRAF6, tumor necrosis factor-α (TNF-α), and NFκB but did not affect MMP-9, nNOS, and nAChR protein.

### 2.5. ACR-Induced TRAF6, IκBα, and NFκB Expression

We conducted the in vitro experiments regarding TRAF6/IκBα/NFκB proteins expression with two cell lines: Neuro-2a and SVG p12. After the Neuro-2a and SVG p12 cells were each treated with 10 μM of ACR for 24 h, TRAF6, IκBα, and NFκB expressions were induced (Figure 7).

## 3. Discussion

To our knowledge, this study was the first to explore the relationship between ACR and depression and the possible role of AQP4 in depression in rats. Some studies have observed changes in brain monoamine levels and depressive symptoms comorbid with anxiety behavior in rodents after exposure to ACR [21,22]. A population-based cross-sectional study evaluated the association between ACR levels in the hemoglobin adducts and depressive symptoms in US adults by using data from the National Health and Nutrition Examination Survey [9].

In our study, the rats treated with ACR for 28 days exhibited symptoms of neurotoxicity, including muscle weakness, difficulty in walking, limb numbness, decreased tactile sensation, and disappearance of tendon reflexes. ACR is toxic to the nervous systems of both humans and animals [7]. It causes swelling at the ends of axons of both the central and peripheral nervous systems and induces paralysis of the cerebrospinal system in humans. Patients presented with severe ACR poisoning experience tremors, ataxia, unconsciousness, dizziness, memory loss, and delusions [8]. ACR also causes nerve palsy and dyskinesia, as well as damage to skeletal muscles, cardiac muscles, and the small intestine [23].

In our study, ACR induced the expression of MMP9 mRNA and MMP-9 proteins in the blood and brain tissues of the rats, respectively. MMP-9 mRNA expression was reported to be significantly higher among patients with depression than in the control group; however, no significant associations between depressive symptoms and MMP-9 mRNA levels were identified [23]. Another study revealed a significant association between MMP-9 serum levels and depression symptoms; the patients’ MMP-9 serum levels decreased after the patients underwent a course of electroconvulsive therapy [24]. MMPs are involved in the pathogenesis of inflammatory diseases, and depression is closely linked to systemic inflammation [25,26]. Several studies have reported associations between depressive symptoms and state-dependent elevations in inflammatory cytokines, such as interleukin-1β, interleukin-6, HO-1, and tumor necrosis factor-α [27,28]. Inflammatory cytokine mediators, such as MMPs, could be pivotal in the clinical progression of depression [29]; modulating inflammatory cytokines, therefore, may ameliorate or lead to the remission of depressive symptoms.

Although AQP4 mRNA was undetected in the blood of the rats in both the ACR-treated groups and the control group, ACR induced the brain AQP4 expression in the ACR-treated groups. Previous clinical studies have revealed that blood mRNA *AQP4* expression in patients with depression was not significantly different from that in the controls [30,31]. In particular, the first aforementioned case–control study [30] identified no differences in blood mRNA *AQP4* expression among responders, patients who were not using drugs, and controls, whereas *AQP4* expression was elevated in patients with treatment-resistant depression. Similarly, the second aforementioned case–control study [31] identified no differences in AQP4 expression in astrocyte-derived extracellular vesicles isolated from the plasma of patients with depression and healthy controls. Overall, this limited evidence seems to suggest AQP4 protein and gene expression levels in the peripheral blood of patients with depression and of healthy individuals differ, indicating that symptoms of ACR-induced neurotoxicity may be related to depression and depressive symptoms.

In the present study, ACR induced blood TRAF6 mRNA and brain TRAF6 protein expression; in addition, NFκB and TNF-α were induced by ACR. Microglia can be classified as either M1 or M2 [32]. When M1 microglia are activated, the expression of proinflammatory cytokines such as inducible nitric oxide synthase (iNOS), TNF-α, and IL-1β increases [33]. When NFkB is activated, pro-inflammatory cytokines such as interleukin-1b (IL-1b), IL-6, and TNF-α are secreted, leading to a neuroinflammatory response. Based on the protein expression results from ACR-treated rats, we used a cell transient transfection model to examine the effects of AQP4 in ACR-induced protein regulation. Cell model results showed AQP4 overexpression would decrease these proteins induced by ACR.

Abundant expression of Toll-like receptor (TLR) 4 induces microglial activation, triggering the TRAF6 [20]. Thus, inhibiting microglial activation is crucial in treating neuroinflammation-associated diseases, including depression. Studies have revealed that TRAF6 plays a key role in NFκB activation [34,35,36]. The results of the present study revealed that ACR exposure triggered IκBα and NFκB expression by mediating the activation of TRAF6 in the cells, indicating that ACR induces the expression of neuroinflammatory mediators regulated by TRAF6–IκBα–NFκB signaling (Figure 8). In this study, the brain AQP4 was decreased, MMP-9 was induced, and TRAF6 pathway triggered in ACR-induced neurotoxicity rats, these phenomena showed the same as depressive symptoms. Transient transfection of AQP4 overexpression could reverse these phenomena induced by ACR. AQP4 plays a very important role in ACR-induced neurotoxicity. The mechanisms underlying the involvement of AQP4 in ACR-induced neurotoxicity and depressive symptoms require further study.

## 4. Materials and Methods

### 4.1. ACR Subacute Oral Toxicity in SD Rats

We obtained 5-week-old male Sprague Dawley (SD) rats from BioLASCO (Taipei, Taiwan). LabDiet 5001 Rodent Diet (PMIR Nutrition International, St. Louis, MO, USA) was selected as feed for the rats, and reverse osmosis water was provided ad libitum. The animals were housed in a room with 40–70% relative humidity, 18–26 °C environmental temperature in a 12 h light–dark cycle. ACR-induced SD rats neurotoxicity dosage of 15 and 30 mg/kg body weight for 28 consecutive days was according to a previous study, as were 15 mg/kg for the moderate dose and 30 mg/kg for the high dose of ACR [37]. ACR was dissolved in drinking water to concentrations of 0, 3, or 6 mg/mL. A dose of 0, 15, or 30 mg/kg ACR in water was administered to the rats by gavage at a rate of 0.5 mL/100 g bw/day. Each dose group consisted of five randomly allocated male rats. After 28 days of treatment, we evaluated walking abnormalities, muscle weakness, and other aspects in rats by a recent study [38]. The rats were euthanized by isoflurane inhalation and exsanguination from the abdominal aorta. Blood samples were collected for gene expression analysis, and their brains were excised for hematoxylin-and-eosin staining or Western blot analysis. To conduct a histopathological evaluation, the rat brains were instilled with 1 mL 10% neutral buffered formalin for at least 1 day. The fixed brains were embedded in paraffin, and 5 µm thick sections were cut from the lobe by microtome (Leica RM2145, Nussloch, Germany) and stained with hematoxylin and eosin. The histopathological score evaluation was performed as previously described [39]. All the animal experiments described in this paper were reviewed and approved by the Institutional Animal Care and Use Committee of National Chung Hsing University (approval number: IACUC No. 107-053).

### 4.2. Blood Gene Expression Difference by ACR in Rats

The total RNA from each rat’s blood sample was extracted using TRIzol Reagent (Invitrogen)/RNeasy Mini Kits (Qiagen, Hilden, Germany). The total RNA of each sample was quantified and qualified using an Agilent 2100 Bioanalyzer (Agilent Technologies, Palo Alto, CA, USA), NanoDrop (Thermo Fisher Scientific, Waltham, MA, USA), and 1% agarose gel. Next-generation sequencing library preparations were constructed using 1 μg total RNA with an RNA integrity number above 6.5, according to the manufacturer’s protocol. The poly(A) mRNA isolation was performed using a poly(A) mRNA Magnetic Isolation Module or rRNA Removal Kit. The mRNA fragmentation and priming were performed using First-Strand Synthesis Reaction Buffer and Random Primers. The first-strand cDNA was synthesized using ProtoScript II Reverse Transcriptase, and the second-strand cDNA was synthesized using Second Strand Synthesis Enzyme Mix. The double-stranded cDNA was purified by beads and treated with End Prep Enzyme Mix for end repair and dA-tailing in one reaction, followed by a T–A ligation to add adaptors to both ends. Size selection of the adaptor-ligated DNA was then performed using beads, and fragments of approximately 420 bp (with insert sizes of approximately 300 bp) were recovered. Each sample then underwent polymerase chain reaction (PCR) amplification for 13 cycles using P5 and P7 primers, both of which carried sequences to enable annealing with flow cells to, in turn, enable bridging PCRs. The P7 primer also carried a 6-base index to allow for multiplexing. The PCR products were cleaned up using beads, validated using a Qsep100 (Bioptic, New Taipei City, Taiwan), and quantified using a Qubit 3.0 fluorometer (Invitrogen, Carlsbad, CA, USA). Libraries with different indices were multiplexed and loaded on an Illumina HiSeq instrument, according to the manufacturer’s instructions (Illumina, San Diego, CA, USA). Sequencing was performed using a 2 × 150-bp paired-end configuration; image analysis and base calling were conducted on the HiSeq instrument with HiSeq Control Software v2.0.12 (Illumina, San Diego, USA) + OLB + GAPipeline-1.6 (Illumina). The sequences were processed and analyzed using GENEWIZ (GeneWiz, Jiangsu, China). Physiological activities involve cooperation among genes with various functions. Pathway functional enrichment facilitates the identification of differentially expressed genes involved in key biochemical metabolic pathways and signal transduction pathways. The KEGG was the primary public pathway database [40] used in this pathway enrichment analysis. In the present study, pathway enrichment analysis was conducted using KEGG pathway units and a hypergeometric test, to determine the pathways of the differentially expressed genes that are significantly enriched against the transcriptome background.

### 4.3. Protein Extraction and Western Blot

The rat brain tissues or harvested cells were placed on ice. The tissues and cells were lysed with cold RIPA lysis buffer (Energenesis Biomedical, Taipei, Taiwan) containing phosphatase inhibitor (MedChemExpress, Monmouth Junction, NJ, USA) and protease inhibitor (Future Scientific, Taoyuan, Taiwan) and shaken for 1 h at 4 °C, to produce homogenates. The homogenates were then moved to Eppendorf tubes and centrifuged at 13,000 rpm at 4 °C. The suspensions were collected as protein lysates. Each quantified protein lysate with loading buffer was heated at 95 °C for 30 min, separated by sodium dodecyl sulfate–polyacrylamide gel electrophoresis with 10% polyacrylamide gel, and electrotransferred to poly(vinylidene fluoride) (PVDF) membranes. The PVDF membranes were then blocked with BlockPro blocking buffer (Visual Protein, Taipei, Taiwan), cut according to the molecular weight of the target protein, and incubated in primary and secondary antibody diluents. The antibodies were diluted with blocking buffer according to the manual instructions. The primary antibodies were rabbit anti-MMP-9 (1:2000; arigo, Hsinchu, Taiwan), AQP4 (1:2000; arigo, Hsinchu, Taiwan), anti-TRAF6, (1:2000; ABclonal, Woburn, MA, USA), anti-nNOS (1:2000; GeneTex, Irvine, CA, USA), anti-NFκB p65 (1:2000; GeneTex, Hsinchu City, Taiwan), anti-IκBα (1:2000; ABclonal, Woburn, MA, USA), anti- and mouse anti-β-actin (1:10,000; Proteintech, Rosemont, IL, USA); secondary antibodies were HRP conjugated goat anti-rabbit IgG (1:10,000; Jackson ImmunoResearch, West Grove, PA, USA) and anti-mouse IgG (1:20,000; arigo, Hsinchu, Taiwan). The expression of each protein was observed with ECL solution (Biokit Biotechnology Incorporation, Miaoli, Taiwan) and FUSION Solo S Chemiluminescence Imaging System (VILBER, Seine-et-Marne, Île-de-France, France).

### 4.4. Cell Culture and ACR Dosage Selection

The mouse neuroblastoma Neuro-2a cells and human glial SVG p12 cells were purchased from the Food Industry Research and Development Institute (Hsinchu, Taiwan). The cells were incubated in Dulbecco’s modified Eagle medium (Thermo Fisher Scientific, Waltham, MA, USA) containing 10% fetal bovine serum (FBS); the Neuro-2a cells were incubated in a minimum essential medium (Thermo Fisher Scientific, Waltham, MA, USA) containing 10% FBS. The cells were cultivated in 10 cm dishes and incubated at 37 °C in a 5% CO_2_ incubator. The number of cells in cell suspension was counted using a cytometer to seed the cells at an appropriate density in each 96-well plate. The cells were seeded in 100 μL culture medium in the triplicate wells of each 96-well plate at a density of 10^3^ cells/well. Once the cells adhered, 50 μL of ACR (0.1–1000 μM) in culture medium was added to each well, and the cells were incubated at 37 °C in a 5% CO_2_ atmosphere. After 24 h of incubation, the solution was removed, and 100 μL of MTT diluted in culture medium to 0.5 mg/mL was added to each well. Subsequently, the cells were incubated for 4 h under the same conditions. The solution in the wells was discarded, and the formazan crystals were dissolved in DMSO (VWR, Radnor, PA, USA). Absorbance at 570 nm was detected using a Multiskan Sky Microplate Reader (Thermo Fisher Scientific, Waltham, MA, USA). The viability of the Neuro-2a cells exposed to ACR for 24 h was evaluated using an MTT assay. The 20% lethal concentration (LC_20_) values for the ACR-exposed Neuro-2a and SVG p12 cells were 11.3 ± 1.6 and 16.2 ± 3.6 μM, respectively. The LC_50_ values for the ACR-exposed Neuro-2a and SVG p12 cells were 28.9 ± 5.0 and 40.3 ± 3.2 μM, respectively. In the MTT assay conducted as part of the in vitro cell model experiment, the ACR concentration was 10 μM. The final concentration of ACR administered to cells was 10 μM.

### 4.5. AQP4 Recombinant Vector Construction and Cell Transient Transfection

The target gene consisted of 906 bp of rat AQP4 cDNA with 5′ end BclI and 3′ end XbaI restriction sites. We constructed the recombinant vectors by digesting mammalian expression vector pcDNA3.0 with restriction enzymes BamHI and XbaI and inserting the target gene by using DNA ligase. The pcDNA3.0-AQP4 recombinant vector was propagated in *Escherichia coli* after construction, and the quality was controlled through DNA sequencing, DNA electrophoresis, and light absorbance. The cells were grown in 6-well plates 1 day before transient transfection and incubated at 37 °C in a 5% CO_2_ atmosphere with a culture medium containing 10% FBS. HyFectTM DNA Transfection Reagent (BioTnA, Kaohsiung, Taiwan) and the DNA vectors were mixed according to the manufacturer’s instructions and reacted for 25 min at room temperature. The culture medium was refreshed prior to transfection. After the transfection reagent and vector mixture were added to the cells, the cells were continuously incubated at 37 °C in a 5% CO_2_ atmosphere, for 24 h. To further investigate the relationship between AQP4 and ACR exposure, we also developed an ACR-exposed transfected cell model. After cells were transfected with pcDNA3.0-AQP4 for 24 h, the culture medium was replaced with a medium containing ACR; the cells were then continuously incubated under the same conditions for another 24 h.

### 4.6. Statistical Analysis

The differences between groups were evaluated through one-way analysis of variance and a two-tailed *t* test conducted using Excel (Microsoft, Redmond, WA, USA). A *p* value of <0.05 was considered significant. The data are presented as means ± standard deviations.

## 5. Conclusions

This study is the first to explore the relationship between ACR-induced neurotoxicity and the role of AQP4 in rats. We determined that ACR induced neurotoxicity by inhibiting AQP4 expression and inducing the TRAF6–IκBα–NFκB pathway in an animal model. Overexpression of AQP4 would reverse this phenomenon in a cell model.

## Figures and Tables

**Figure 1 molecules-27-01066-f001:**
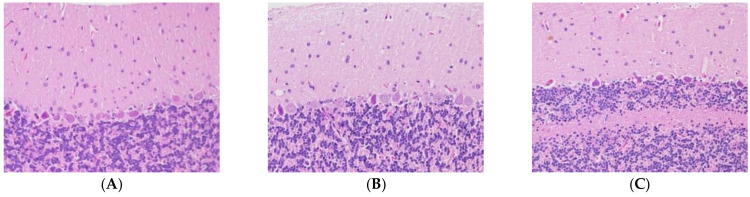
Brain neurons in the molecular, Purkinje cell, and granular layers: (**A**) control group, (**B**) 15 mg/kg acrylamide (ACR) group, and (**C**) 30 mg/kg ACR group.

**Figure 2 molecules-27-01066-f002:**
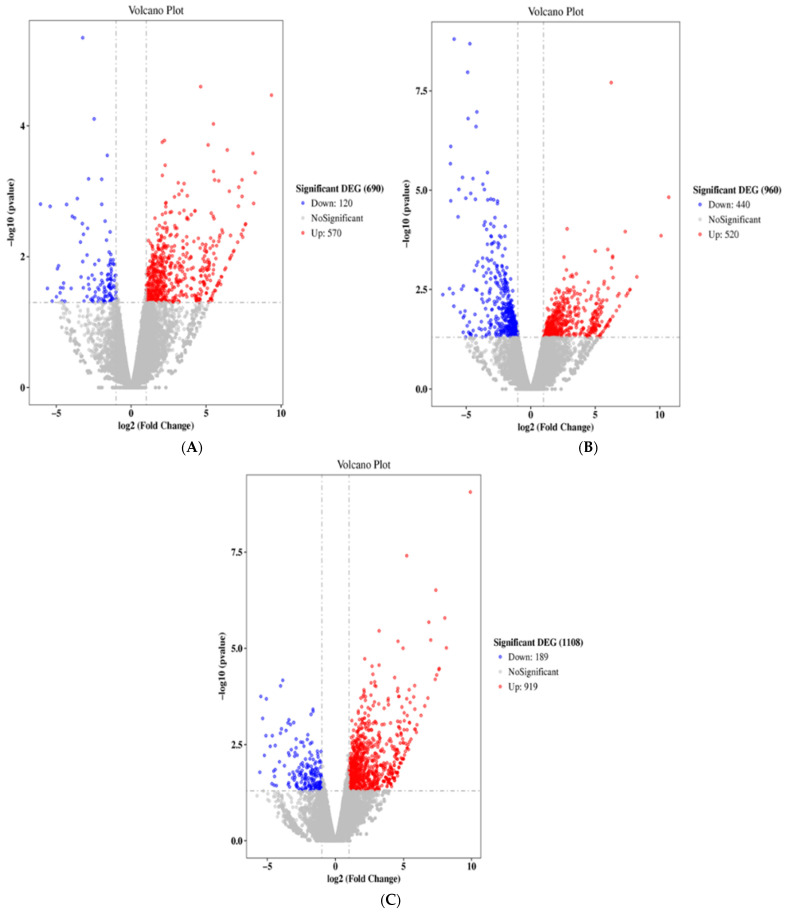
Volcano plots of differential expression. Red and blue dots represent significantly upregulated and significantly downregulated genes, respectively. X axis: log2 fold change in gene expression. Y axis: statistical significance of the differential expression in log10 (*p* value): (**A**) 15 mg/kg ACR group versus control group; (**B**) 30 mg/kg ACR group versus the control group; (**C**) 15 mg/kg and 30 mg/kg groups versus the control group.

**Figure 3 molecules-27-01066-f003:**
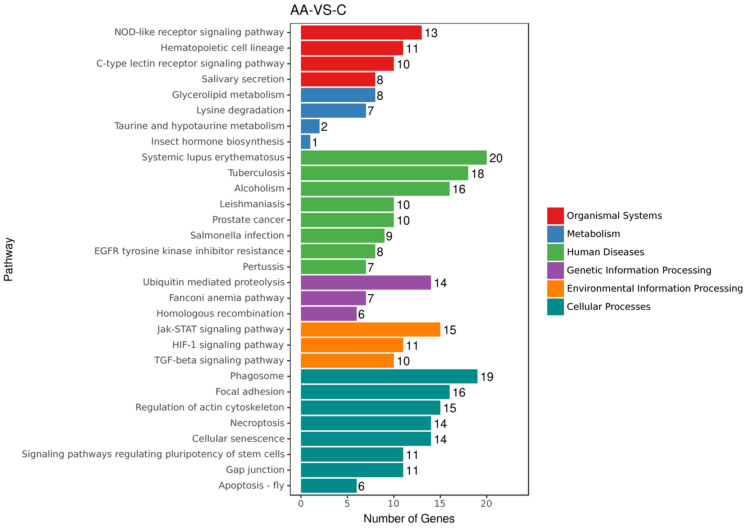
Kyoto Encyclopedia of Genes and Genomes enrichment histogram. X axis: gene number. Y axis: pathway term. There were six pathways difference between control and ACR-treated groups—namely, organismal systems, metabolism, human disease, genetic information processing, environmental information processing, and cellular processes.

**Figure 4 molecules-27-01066-f004:**
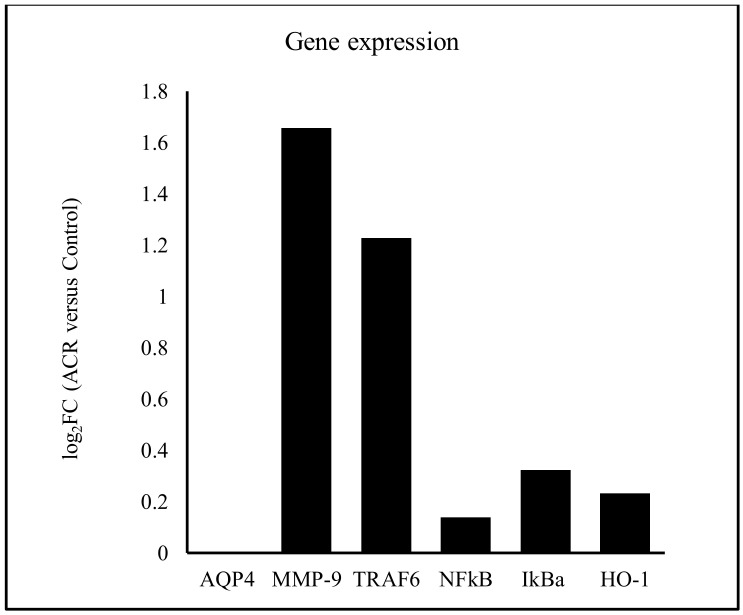
Relative *AQP4*, *MMP-9*, *TRAF6*, *NF**κB*, *I**κB**α*, and *HO-1* mRNA expression in the blood of the rats: 15 mg/kg and 30 mg/kg ACR groups versus control group.

**Figure 5 molecules-27-01066-f005:**
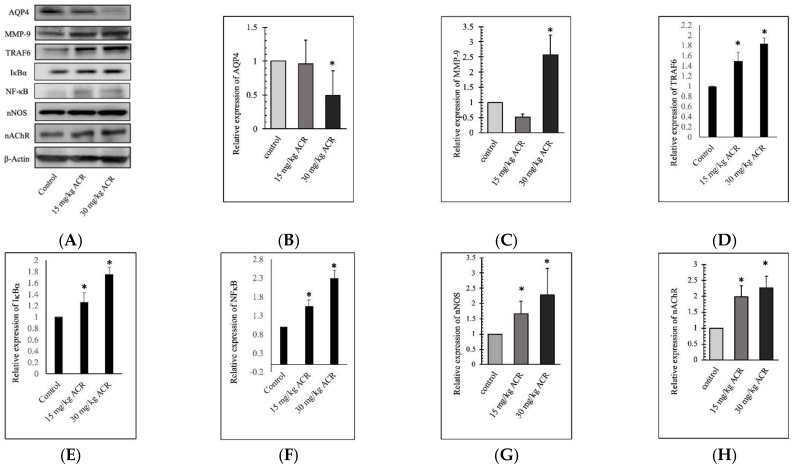
Protein expression in the brain tissues of rats treated with 0, 15, or 30 mg/kg ACR for 28 days. Brain tissue lysates were used for the Western blot analysis. ACR inhibited AQP4 expression and induced MMP-9, TRAF6, IκBα, NFκB, nNOS, and nAChR expression (**A**). The quantitative data showing AQP4 (**B**), MMP-9 (**C**), TRAF6 (**D**), IκBα (**E**), NFκB (**F**), nNOS (**G**), and nAChR (**H**) protein levels. *: Significant difference versus the control group.

**Figure 6 molecules-27-01066-f006:**
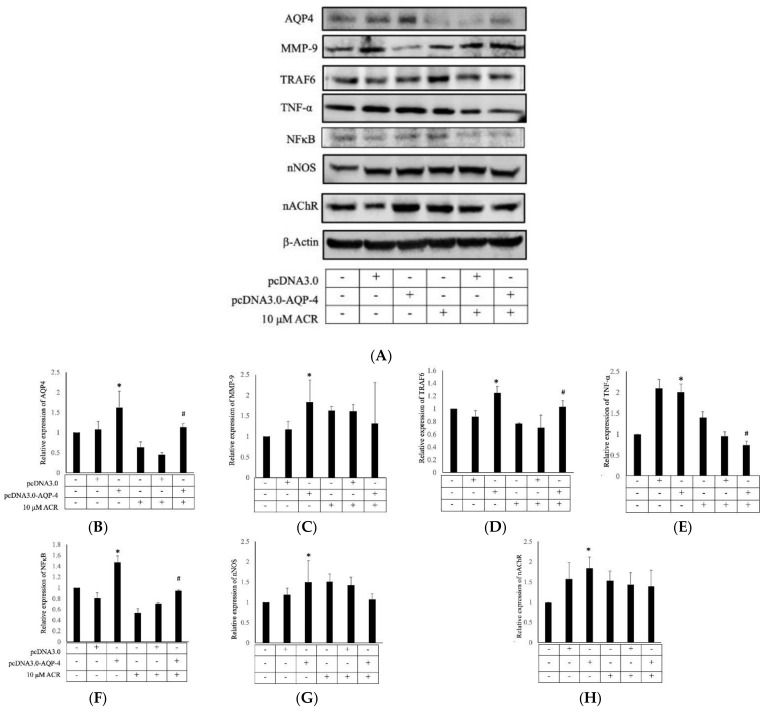
Transient transfection of AQP4 regulated MMP-9 and TRAF6 expression. Neuro-2a cells were transiently transfected with pcDNA3.0-AQP4 for 48 h. After transfection for 24 h, 10 μM of ACR was added for the remaining 24 h. Transient transfection of AQP4 inhibited MMP-9, TRAF6, and NFκB expression but did not affect nNOS or nAChR expression (**A**). In the ACR-treated groups, ACR inhibited AQP4 expression and inhibited TRAF6 and TNF-α expression. The quantitative data showing AQP4 (**B**), MMP-9 (**C**), TRAF6 (**D**), TNF-α (**E**), NFκB (**F**), nNOS (**G**), and nAChR (**H**) protein levels. *: Significant difference versus the control group, #: Significant difference versus the ACR treatment group.

**Figure 7 molecules-27-01066-f007:**
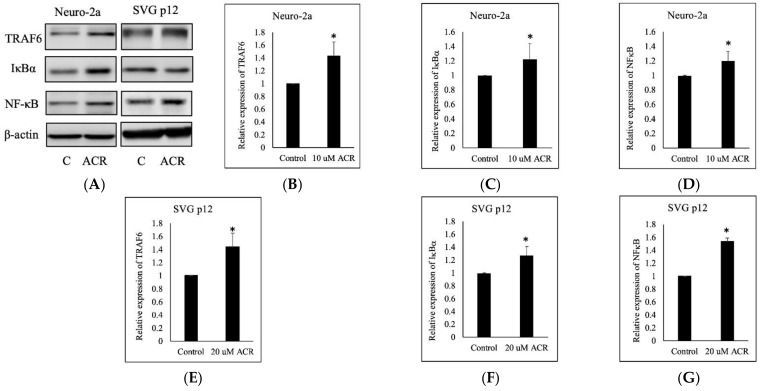
Effect of ACR on the TRAF6–NFκB pathway. Neuro-2a and SVG p12 cells were treated with 10 μM of ACR for 24 h. ACR induced TRAF6 overexpression and IκBα and NFκB expression in both types of cells (**A**). The quantitative data showing Neuro-2a cell of TRAF6 (**B**), IκBα (**C**), NFκB (**D**), and SVG p12 cell of TRAF6 (**E**), IκBα (**F**), NFκB (**G**) protein levels. *: Significant difference versus the control group.

**Figure 8 molecules-27-01066-f008:**
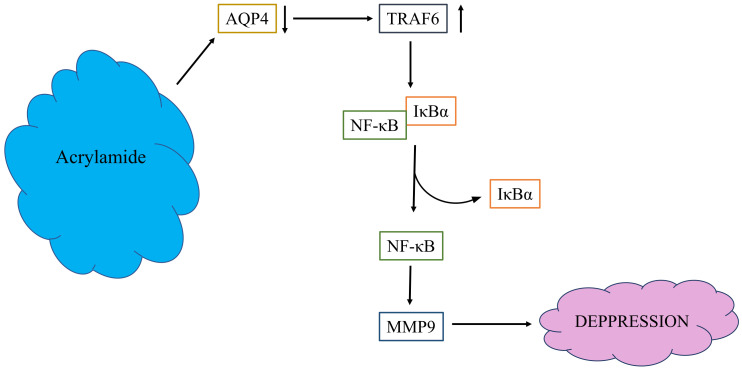
The possible pathway media by ACR.

**Table 1 molecules-27-01066-t001:** Relative gene expression associated with depressive symptoms and ROS.

Gene ID	Gene Name	Log_2_FC	*p*-Value
ENSRNOG00000016043	*AQP4*	N/A	N/A
ENSRNOG00000017539	*MMP-9*	1.65759	0.088937
ENSRNOG00000016606	*TRAF6*	1.227902	0.028015
ENSRNOG00000023258	*NF* *κB*	0.13844	0.649575
ENSRNOG00000007390	*I* *κB* *α*	0.32284	0.411419
ENSRNOG00000003773	*HO-1*	0.231468	0.712061

## Data Availability

Not available.

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
