# Peer review of "AQP4 Attenuated TRAF6/NFκB Activation in Acrylamide-Induced Neurotoxicity"

_molecules, 2022, doi:10.3390/molecules27031066_

Round 1
Reviewer 1 Report
Review Report
- A brief summary
The present article titled “AQP4 attenuated TRAF6/NFB activation in acrylamide-induced neurotoxicity” is focused on the mechanisms underlying ACR-induced neurotoxicity associated with ROS activation, depressive symptoms, and induction of MMP-9 and AQP4 protein expression in rat brains and in vitro study.
- Comments
Introduction
-The authors should look at the sentence “Excess proinflammatory cytosines have….” for errors.
Materials and Methods
-The authors should explain the meaning of IC20 and its relation to the cytotoxicity of ACR
Results
Acrylamide-induced neurotoxicity symptoms in rats -Please specify the meaning of the following sentence: “The pathological exam results revealed no differences in the the neurons in the molecular, Purkinje cell, 90 and granular layers”. Like that the sentence does not correspond with the next sentence's meaning.
Author Response
A brief summary
The present article titled “AQP4 attenuated TRAF6/NFB activation in acrylamide-induced neurotoxicity” is focused on the mechanisms underlying ACR-induced neurotoxicity associated with ROS activation, depressive symptoms, and induction of MMP-9 and AQP4 protein expression in rat brains and in vitro study.
Comments
Introduction
-The authors should look at the sentence “Excess proinflammatory cytosines have….” for errors.
Response: Thank you very much for your professional comments, according to your comments we revised our manuscript. We revised the sentence “Excess proinflammatory cytokines have…” on line 60.
Materials and Methods
-The authors should explain the meaning of IC20 and its relation to the cytotoxicity of ACR
Response: We have revised the sentence: “The 20% lethal concentration (LC20) values for the ACR-exposed Neuro-2a and SVG p12 cells were 11.3 ± 1.6 and 16.2 ± 3.6 μM, respectively.” for the meaning of cytotoxicity of ACR. Line 348-349
Results
Acrylamide-induced neurotoxicity symptoms in rats -Please specify the meaning of the following sentence: “The pathological exam results revealed no differences in the the neurons in the molecular, Purkinje cell, 90 and granular layers”. Like that the sentence does not correspond with the next sentence's meaning.
Response: We have revised the sentence as “The pathological exam results revealed the normal morphologies of neurons in the molecular, Purkinje cell, and granular layers in the control group (Figure 1).” to correspond with the next sentence’s meaning. Line 89-91.
Reviewer 2 Report
- Acrylamide treatment has been already investigated, this research is not particularly innovative, and the tested signaling pathways have been involved in many processes, while the regulation between the examined parameters has not been described in detail.
- It is necessary to state how the selected doses for the in vivo study and the time of application of the treatment in the study were determined, and more information on whether any previous experiments were performed or the doses and times were selected based on literature.
- Positive control is missing.
- Gene expression of brain tissue is missing.
- In addition to the Western blot method, add immunofluorescent method to confirm the presence and location of the protein.
- Regarding Neuro2 and SVGp12, it is not stated which cell lines are, why they were selected and how they differ, as well as the concentrations selected why they were selected and how the treatment time was selected.
- Data on cytotoxicity of treatment are missing.
- Part of the discussion and conclusion should be improved.
- It is necessary to reduce the number of references and introduce references of a more recent date.
Author Response
1. Acrylamide treatment has been already investigated, this research is not particularly innovative, and the tested signaling pathways have been involved in many processes, while the regulation between the examined parameters has not been described in detail.
Response: Thank you very much for your professional comments, according to your comments we revised our manuscript. We added the sentence ”According to the Kyoto Encyclopedia of Genes and Genomes (KEGG) public pathway database, there were six pathways difference between control and ACR-treated groups, including organismal systems, metabolism, human disease, genetic information processing, environmental information processing, and cellular processes (Figure 3).” to describe them in detail, on line 125-126.
2. It is necessary to state how the selected doses for the in vivo study and the time of application of the treatment in the study were determined, and more information on whether any previous experiments were performed or the doses and times were selected based on literature.
Response: We added the sentence on line 222: “ACR-induced SD rats neurotoxicity dosage of 15 and 30 mg/kg body weight for 28 consecutive days was according to a previous study, also, the 15 mg/kg for the moderate-dose and 30 mg/kg for high-dose of ACR [37].” To explain the selected doses for the in vivo study and the time of the treatment in the study were determined. And cited a reference 37. Line 255-257.
3. Positive control is missing.
Response: In the in vivo study, we used control group, 15 mg/kg ACR group, and 30 mg/kg ACR group, there were no positive control groups since we were looking for the effects of ACR on gene expression in peripheral blood and protein expression in the brain protein lysis. In the cell model, we used ACR to treat cells or transient transfection to ensure the possible pathway from in vivo study results. We did not use positive control groups in this study.
4. Gene expression of brain tissue is missing.
Response: Thank you very much for the suggestion. In this study, we used peripheral blood mRNA expression difference to predict the ACR-induced neurotoxicity, we used SD rat’s brain lysis to analyze the protein expression to research the possible mechanism of neurotoxicity. We did not use brain tissue to be evaluated the gene expression, since the experimental rats’ brains were divided into 2 sections, one for pathological examination, and the other for protein lysis. Maybe we will try to performance the brain gene expression in the next study.
5. In addition to the Western blot method, add immunofluorescent method to confirm the presence and location of the protein.
Response: Thank you very much for the suggestion. In our study, the expression of AQP4 expression was inhibited by ACR in the brain tissue, the brain cells co-expression of associated protein may be hard to ensure the correlation. But, we will try to perform the immunofluorescent method in the cell model in the next study.
6. Regarding Neuro2 and SVGp12, it is not stated which cell lines are, why they were selected and how they differ, as well as the concentrations selected why they were selected and how the treatment time was selected.
Response: We revised the first sentence of 4.4 section: “The mouse neuroblastoma Neuro-2a cells and human glial SVG p12 cells were purchased from the Food Industry Research and Development Institute (Hsinchu, Taiwan). (line 331)” to state the cell lines are and how they differ. The viability of the Neuro-2a cells exposed to ACR for 24 h was evaluated using an MTT assay. We revised as: The 20% lethal concentration (LC20) values for the ACR-exposed Neuro-2a and SVG p12 cells were 11.3 ± 1.6 and 16.2 ± 3.6 μM, respectively. The LC50 value for the ACR-exposed Neuro-2a and SVG p12 cells were 28.9 ± 5.0 and 40.3 ± 3.2 μM, respectively. In the MTT assay conducted as part of the in vitro cell model experiment, the ACR concentration was 10 μM. The final concentration of ACR administered to cells was 10 μM. (Line 348-352)
7. Data on cytotoxicity of treatment are missing.
Response: The 20% lethal concentration (LC20) values for the ACR-exposed Neuro-2a and SVG p12 cells were 11.3 ± 1.6 and 16.2 ± 3.6 μM, respectively. The LC50 value for the ACR-exposed Neuro-2a and SVG p12 cells were 28.9 ± 5.0 and 40.3 ± 3.2 μM, respectively. In the MTT assay conducted as part of the in vitro cell model experiment, the ACR concentration was 10 μM. The final concentration of ACR administered to cells was 10 μM. (Line 348-352)
8. Part of the discussion and conclusion should be improved.
Response: We revised the discussion and conclusion to improve the manuscript.
9. It is necessary to reduce the number of references and introduce references of a more recent date.
Response: We reduce the number of references to 40, and added the more recent reference in the introduction.
Reviewer 3 Report
The manuscript presents data concerning neurotoxicity from chronic exposure to acrylamide. Some points need to be clarified to give a coherent and unified vision of the results.
The relationship between in vivo and in vitro experiments must be clarified, especially at the level of the purpose of the research and of the discussion/conclusion. What is the reason for the choice of those two cell lines? Only the neuro-2a line was transfected, why? Why was AQP4 mRNA not determined in the blood? Is the value too low, below the sensitivity limits? On line 185 the authors write that ACR induced the AQPR4 expression, are they referring to the brain? Can the authors make some assumptions about the mechanisms that underlying the involvement of AQP4 in ACR-induced neurotoxicity?
The verb in the sentence is missing "Patients with severe ACR poisoning experience tremors, ataxia, unconsciousness, dizziness, memory loss, and delusions [17]."
In the Materials and Methods the authors write on line 220 that the relative humidity was 40-70% at constant 18-26°C environmental temperature, but in my opinion they are very wide ranges, we cannot speak of constancy.
Substantially it is necessary that the authors clearly specify the objectives of the different experimental approaches, what they expected to obtain and that they relate the results obtained with the two experiments
Author Response
The manuscript presents data concerning neurotoxicity from chronic exposure to acrylamide. Some points need to be clarified to give a coherent and unified vision of the results.
Response: Thank you very much for your professional comments, according to your comments we revised our manuscript.
The relationship between in vivo and in vitro experiments must be clarified, especially at the level of the purpose of the research and of the discussion/conclusion. What is the reason for the choice of those two cell lines? Only the neuro-2a line was transfected, why? Why was AQP4 mRNA not determined in the blood? Is the value too low, below the sensitivity limits? On line 185 the authors write that ACR induced the AQPR4 expression, are they referring to the brain? Can the authors make some assumptions about the mechanisms that underlying the involvement of AQP4 in ACR-induced neurotoxicity?
Response:
- We revise the relationship between in vivo and in vitro experiments in the discussion section, lines 225-228: According to the protein expression results from ACR-treated rats, we used a cell transient transfection model to examine the effects of AQP4 in ACR-induced protein regulation. Cell model results showed AQP4 overexpression would decrease these proteins induced by ACR.
- We chose the mouse neuroblastoma Neuro-2a cells and human glial SVG p12 cells to show 2 different species cell lines, one for mouse and one for human, and compare to in vivo rat’s result.
- Only the neuro-2a cell was transfected since the AQP4 plasmid was designed from the mouse gene.
- Yes, blood AQP4 mRNA level was too low to be detected by RNAseq method.
- Yes, we revised and added ‘the brain’ in line 212.
- We revised the mechanisms as: “In this study, brain AQP4 was decreased, MMP-9 was an induction, and TRAF6 pathway triggered in ACR-induced neurotoxicity rats, these phenomena showed the same as depressive symptoms. Transient transfection of AQP4 overexpression could reverse these phenomena induced by ACR. AQP4 plays a very important role in ACR-induced neurotoxicity.” In line 240-244.
The verb in the sentence is missing "Patients with severe ACR poisoning experience tremors, ataxia, unconsciousness, dizziness, memory loss, and delusions [17]."
Response: We revised the sentence as “Patients showed with severe ACR poisoning experience tremors, ataxia, unconsciousness, dizziness, memory loss, and delusions.” In line 193.
In the Materials and Methods the authors write on line 220 that the relative humidity was 40-70% at constant 18-26°C environmental temperature, but in my opinion they are very wide ranges, we cannot speak of constancy.
Response: We have deleted “at constant”.
Substantially it is necessary that the authors clearly specify the objectives of the different experimental approaches, what they expected to obtain and that they relate the results obtained with the two experiments
Response: We revised the sentences in lines 96-97, 100, 117-118, 133, 136-138, and 167-168.
Reviewer 4 Report
In this manuscript, the authors studied the effect of acrylamide (ACR) toxicity by repetitive feeding ACR (0, 15 and 30 mg/kg) to rats. mRNA levels from blood samples and protein expression from brain samples were analyzed. The authors stated that ACR caused neurotoxicity in rats and Purkinje cells abnormalities; however, these results do not seem clear. Moreover, they found that ACR induced MMP-9, TRAF6, NFkB, IkBa expression (mRNA levels in blood samples and protein in brain tissue), and inhibited AQP4 protein levels in brain (AQP4 mRNA was not detected in blood). Moreover, AQP4 overexpression in cultured cells inhibited MMP-9, TRAF6, and NFkB expression. But analysis of protein levels lacks quantitative information.
The manuscript aims to provide relevant molecular findings for the relationship between ACR and depression and the possible role of AQP4 in depression. However, there are some results and some methodological aspects that need to be clarified, quantitative data need to be included, and the discussion of some results can be improved. Thus, I suggest that authors provide further details and revise some issues as described below.
- May the authors provide additional information and justification on the methodology used, as so: a) Why they use only high concentrations of 15 and 30 mg/kg of ACR? b) How they evaluated walking abnormalities, muscle weakness and other aspects in rats, described in the results? c) Why did authors analyzed mRNA levels from blood and not also from brain samples?
- Line 87: “Rats treated with 30 mg/kg ACR exhibited hind-limb muscle weakness…”. Did the authors evaluate the same aspects in rats treated with 15 mg/kg ACR? There were any differences between the 15 mg/kg ACR treated rats and controls?
- Line 89-94: “The pathological exam results revealed no differences in the the neurons in the molecular, Purkinje cell…The Purkinje cells of the rats in the 15 mg/kg and 30 mg/kg ACR groups exhibited abnormal morphologies…”. a) May be the authors clarify if cerebellar cells (Purkinje cells and others) are altered in ACR treated rats? b) Did the authors analyzed neuronal cells in other brain regions besides cerebellum?
- Line 125-126 and Figure 5. a) Can the authors indicate which brain area was analyzed? (cortex, cerebellum?) b) Quantitative data showing the protein levels (from different rat brains) is lacking and it is essential for taking conclusions.
- Line 140, Figure 6. Quantitative data is lacking, which is also important to infer about ACR effect on protein levels that are not clear in the figures.
- Line 150, Figure 7: Quantitative data showing the protein levels (from different rat brains) is lacking.
I also have some minor comments:
- Line 22-23, abstract: “..which subsequently presented with depressive symptoms.” Did the authors evaluate depressive symptoms?
- Line 60: “citosines”. Minor typo - cytokines.
- Line 67: “…of MDDAquaporin 4 (AQP4)…”. Minor typo, the words are linked.
- Line 111, Figure 2. The quality of the image does not allow to read the text.
- Line 117, Figure 3. The legend of figure 3 needs additional information.
- Line 213, Figure 8. There is no reference to figure 8 in the text
- Line 301/302. The final concentration of ACR administered to cells should be indicated.
Author Response
In this manuscript, the authors studied the effect of acrylamide (ACR) toxicity by repetitive feeding ACR (0, 15 and 30 mg/kg) to rats. mRNA levels from blood samples and protein expression from brain samples were analyzed. The authors stated that ACR caused neurotoxicity in rats and Purkinje cells abnormalities; however, these results do not seem clear. Moreover, they found that ACR induced MMP-9, TRAF6, NFkB, IkBa expression (mRNA levels in blood samples and protein in brain tissue), and inhibited AQP4 protein levels in brain (AQP4 mRNA was not detected in blood). Moreover, AQP4 overexpression in cultured cells inhibited MMP-9, TRAF6, and NFkB expression. But analysis of protein levels lacks quantitative information.
The manuscript aims to provide relevant molecular findings for the relationship between ACR and depression and the possible role of AQP4 in depression. However, there are some results and some methodological aspects that need to be clarified, quantitative data need to be included, and the discussion of some results can be improved. Thus, I suggest that authors provide further details and revise some issues as described below.
1. May the authors provide additional information and justification on the methodology used, as so: a) Why they use only high concentrations of 15 and 30 mg/kg of ACR? b) How they evaluated walking abnormalities, muscle weakness and other aspects in rats, described in the results? c) Why did authors analyzed mRNA levels from blood and not also from brain samples?
Response: Thank you very much for your professional comments, according to your comments we revised the manuscript. a). According to reference 37, we use high concentrations of 15 and 30 mg/kg of ACR in the subacute oral toxicity test. We have revised lines 255-257. b). We evaluated walking abnormalities, muscle weakness, and other aspects in rats by a recent study [38], and we revised in line 261-262. Although we did not record the gait score, it showed typical neurotoxicity symptoms. c). In this study, we used peripheral blood mRNA expression difference to predict the ACR-induced neurotoxicity, we used SD rat’s brain lysis to analyze the protein expression to research the possible mechanism of neurotoxicity. We did not use brain tissue to be evaluated the gene expression, since the experimental rats’ brains were divided into 2 sections, one for pathological examination, and the other for protein lysis.
2. Line 87: “Rats treated with 30 mg/kg ACR exhibited hind-limb muscle weakness…”. Did the authors evaluate the same aspects in rats treated with 15 mg/kg ACR? There were any differences between the 15 mg/kg ACR treated rats and controls?
Response: Yes, there was a difference between the 15 mg/kg ACR treated rats and controls. We revised section 2.1, lines 86-87: Rats treated with 15 mg/kg ACR showed hind-limb and some difficulty in walking.
3. Line 89-94: “The pathological exam results revealed no differences in the the neurons in the molecular, Purkinje cell…The Purkinje cells of the rats in the 15 mg/kg and 30 mg/kg ACR groups exhibited abnormal morphologies…”. a) May be the authors clarify if cerebellar cells (Purkinje cells and others) are altered in ACR treated rats? b) Did the authors analyzed neuronal cells in other brain regions besides cerebellum?
Response: a). We have revised the sentence as “The pathological exam results revealed the normal morphologies of neurons in the molecular, Purkinje cell, and granular layers in the control group (Figure 1).” (lines 89-91) to correspond with the next sentence’s meaning. The Purkinje cells of the rats in the 15 mg/kg and 30 mg/kg ACR groups exhibited abnormal morphologies; the Purkinje cells in the cerebellar cortices of the rats had irregular shapes and elevated basophilic chromatin, which manifested as deep-stained nuclei with no visible nucleoli. The surrounding tissue gap became wider. b). We analyzed neuronal cells in the other brain regions besides the cerebellum, there was no gross finding by pathological exam.
4. Line 125-126 and Figure 5. a) Can the authors indicate which brain area was analyzed? (cortex, cerebellum?) b) Quantitative data showing the protein levels (from different rat brains) is lacking and it is essential for taking conclusions.
Response: a). The protein lysis was extracted by left total brain tissue. b). We added the quantitative data in Figure 5 B-H.
5. Line 140, Figure 6. Quantitative data is lacking, which is also important to infer about ACR effect on protein levels that are not clear in the figures.
Response: We added the quantitative data in Figure 6 B-H.
6. Line 150, Figure 7: Quantitative data showing the protein levels (from different rat brains) is lacking.
Response: We added the quantitative data in Figure 7 B-G.
I also have some minor comments:
7. Line 22-23, abstract: “..which subsequently presented with depressive symptoms.” Did the authors evaluate depressive symptoms?
Response: We revised it as “..which subsequently presented with neurotoxicity.”
8. Line 60: “citosines”. Minor typo - cytokines.
Response: We revised it as “cytokines.”
9. Line 67: “…of MDDAquaporin 4 (AQP4)…”. Minor typo, the words are linked.
Response: We revised it as MDD Aquaporin (AQP4).
10. Line 111, Figure 2. The quality of the image does not allow to read the text.
Response: We revised new photos for allowing to read.
11. Line 117, Figure 3. The legend of figure 3 needs additional information.
Response: We added the sentence: There were six pathways difference between control and ACR-treated groups, including organismal systems, metabolism, human disease, genetic information processing, environmental information processing, and cellular processes
12. Line 213, Figure 8. There is no reference to figure 8 in the text
Response: We added the reference to figure 8 in line 240.
13. Line 301/302. The final concentration of ACR administered to cells should be indicated.
Response: We revised it as: The final concentration of ACR administered to cells was 10 μM. In line 252.
Round 2
Reviewer 2 Report
There are some typographical errors so, please read the text and correct it.
Author Response
There are some typographical errors so, please read the text and correct it.
Response:
Thank you very much for your comments, according to your comments we revised our manuscript.
Reviewer 3 Report
The manuscript has been revised and I believe it can now be published
Author Response
The manuscript has been revised and I believe it can now be published
Response:
Thank you very much for your comments.
Reviewer 4 Report
The authors have satisfactorily addressed the comments raised in the previous round of review. But I still have some minor revision points, including the rewriting of some sentences:
- Lines 134-136: The authors refer that ACR treatment induced “dose-dependent” increases in MMP-9 and other proteins, however looking at the graph, treatment with 15mg/Kg does not affect MMP-9 expression. The same happens with AQP4; treatment with 15mg/Kg does not affect AQP4 expression. Thus, the authors should clarify this in the text.
- Several added sentences should be revised for typos and spell check:
- Line 87
- Line 95
- Line 100
- Lines 111-112
- Lines 117-118
- Line 125
- Line 133
- Line 137
- Lines 146, 165-166, 179 (“significant difference versus control”)
- Line 242
Author Response
The authors have satisfactorily addressed the comments raised in the previous round of review. But I still have some minor revision points, including the rewriting of some sentences:
1. Lines 134-136: The authors refer that ACR treatment induced “dose-dependent” increases in MMP-9 and other proteins, however looking at the graph, treatment with 15mg/Kg does not affect MMP-9 expression. The same happens with AQP4; treatment with 15mg/Kg does not affect AQP4 expression. Thus, the authors should clarify this in the text.
Response:
Thank you very much for your comments, according to your comments, we revised our manuscript. We revised the sentence as “The 28-day ACR treatment-induced MMP-9 overexpression in the 30 mg/kg group and dose-dependent increases in TRAF6, IkBa, NFkB, nNOS, and nAChR expression (Figure 5). However, ACR inhibited AQP4 expression in the 30 mg/kg group.”
2. Several added sentences should be revised for typos and spell check:
1. Line 87
Response: We revise it as “rats”.
2. Line 95
Response: We revised it as “indicate”.
3. Line 100
Response: We revised it as “administered oral”.
4. Lines 111-112
Response: We revised it as “Fanconi anemia pathway”
5. Lines 117-118
Response: We revise it as “indicate”.
6. Line 125
Response: We added the period.
7. Line 133
Response: We revised the sentences as “The 28-day ACR treatment-induced MMP-9 overexpression in the 30 mg/kg group and dose-dependent increases in TRAF6, IkBa, NFkB, nNOS, and nAChR expression (Figure 5). However, ACR inhibited AQP4 expression in the 30 mg/kg group.”
8. Line 137
Response: We revised the sentence as “Results showed ACR-induced depressive symptoms associated with protein MMP-9 activation and AQP4 inhibition.”
9. Lines 146, 165-166, 179 (“significant difference versus control”)
Response: We revised them as “Significant difference versus the control group.” In line 145, “*: Significant difference versus the control group, #: Significant difference versus the ACR treatment group.” In lines 164-165, and “*: Significant difference versus the control group.” In line 178.”
10. Line 242
Response: We revise it as “induced” in line 241.